# Characteristics of pediatric emergency department frequent visitors and their risk of a return visit: A large observational study using electronic health record data

**Sanne E. W. Vrijlandt**[1], **Daan Nieboer**[2], **Joany M. Zachariasse**[1], **Rianne Oostenbrink**[1]*

**1** Department of General Pediatrics, Erasmus MC-Sophia Children's Hospital, Rotterdam, The Netherlands,
**2** Center for Medical Decision Sciences, Department of Public Health, Erasmus MC, Rotterdam, The Netherlands

* r.oostenbrink@erasmusmc.nl

## Abstract

### Background

Among pediatric emergency department (ED) visits, a subgroup of children repeatedly visits the ED, making them frequent visitors (FVs). The aim of this study is to get insight into the group of pediatric ED FVs and to determine risk factors associated with a revisit.

### Methods and findings

Data of all children aged 0–18 years visiting the ED of a university hospital in the Netherlands between 2017 and 2020 were included in this observational study based on routine data extraction. Children with 4 or more ED visits within 365 days were classified as FVs. Descriptive analysis of the study cohort at patient- and visit-level were performed. Risk factors for a recurrent ED visit were determined using a Prentice Williams and Peterson gap time cox-based model. Our study population of 10,209 children with 16,397 ED visits contained 500 FVs (4.9%) accounting for 3,481 visits (21.2%). At patient-level, FVs were younger and more often suffered from chronic diseases (CDs). At visit-level, frequent visits were more often initiated by self-referral and were more often related to medical problems (compared to trauma's). Overall, FVs presented at the ED more often because of an infection (41.3%) compared to non-FVs (27.4%), either associated or not with the body system affected by the CD. We identified the presence of a comorbidity (non-complex CD HR 1.66; 1.52–1.81 and complex CD HR 2.00; 1.84–2.16) as determinants with the highest hazard for a return visit.

### Conclusion

Pediatric ED FVs are a small group of children but account for a large amount of the total ED visits. FVs are younger patients, suffering from (complex) comorbidities and present more often with infectious conditions compared to non-FVs. Healthcare pathways, including safety-netting strategies for acute manifestations from their comorbidity, or for infectious

**Data Availability Statement:** A minimum anonymized database to reproduce the analyses is

available through the supporting information file 6. If these data are used for other research questions, users are requested to notify the senior author (r. oostenbrink@erasmusmc.nl) or by contacting PlosOne.

**Funding:** The funders had no role in study design, data collection and analysis, decision to publish, or preparation of the manuscript.

**Competing interests:** The authors have declared that no competing interests exist.

conditions in general may contribute to support parents and redirect some patients from the ED.

## Introduction

The emergency department (ED) is an important location where patients of all ages receive acute medical care. Patients can either visit the ED at their own initiative (self-referral); by out of hospital emergency services or referred by a previous (emergent) health care contact from either primary or secondary care. In the Netherlands, the ED visiting rate is reported to be 115 visits per 1000 inhabitants [1] with 1 out of 5 below 18 years old or younger [2].

Of all ED visitors, a subgroup of patients frequently seeks medical attention at the ED. Therefore, a small group of patients accounts for a large number of visits, making them frequent visitors (FV). The high number of visits made by FVs may partially contribute to ED crowding [3–5].

Apart from ED overload, visiting the ED is an expensive form of receiving care. In the Netherlands, visiting an ED is 2.5 times as expensive as visiting a pediatrician and almost 8 times as expensive as visiting the GP [6]. It was found that 31% of the cost of all pediatric ED patients was made by the FVs consisting of only 8% of the patients [7]. Therefore, preventing repeated visits by FVs or channeling FVs to another healthcare path might also reduce total health care costs. In addition, visiting a familiar doctor might improve patient satisfaction and patient outcomes [8].

It is suggested that the visits by FVs may be unnecessary improving patient education and developing safety netting pathways might help to reduce these visits. However, redirecting care from the ED, or providing fast tracks for specific patient groups might also influence necessary visits and quality of care.

Insight in the characteristics of pediatric FVs may assist in optimizing care before and during the ED visit, as it might be possible to fulfill their medical needs in other healthcare pathways. Several studies in adults have identified that FVs were more often patients with pre-existing comorbidities [9]. Studies in pediatric ED populations have been limited. These studies showed FVs were associated with a younger age, a lower socioeconomic status and having chronic conditions. Asthma, infectious ear/nose/sinus disorders and other respiratory disorders were the most common diagnoses in these FVs at the pediatric ED [10, 11]. However, most pediatric studies were performed in the US with a different healthcare system and most did not include both visit and patient characteristics.

Therefore, the aim of this study is to get insight into the group of pediatric FVs and to determine risk factors associated with a revisit to the ED. This contributes to more comprehensive information about FV's at the ED for healthcare professionals and policy makers. As a result, FVs healthcare pathways for acute conditions can be optimized, for example by channeling visits and safety-netting strategies.

## Methods

### Study design, patient selection and setting

This observational study was based on automatic data extraction of the electronic medical record system including all pediatric patients visiting the ED of the Erasmus MC-Sophia Children's Hospital (Rotterdam, The Netherlands) between the first of July 2017 and the 31st of December 2020. The Erasmus MC is a large inner-city academic hospital. The

Erasmus MC-Sophia Children's Hospital provides general pediatric care for the inner city of Rotterdam. In addition, it provides tertiary care for the South/Southwest region of the Netherlands. The cohort consists of all patients 0–18 years who presented at the ED of the Erasmus MC-Sophia Children's Hospital within the study period. The dataset was anonymized and the study was approved by the medical ethical committee and informed consent was waived.

## Definition of frequent visitors and frequent visits

In line with previous literature, at patient level a patient who visited the ED at least 4 times within 365 days, starting from the index visit, was categorized as a FV [12–16]. The first visit of the sequence of 4 or more visits within 365 days was marked as the index visit. At visit level, visits with 3 other visits within 365 days (either before or after the visit) are categorized as frequent visits (Fig 1).

## Data collection

Data were extracted from the electronic hospital information system from routinely documented data by trained nurses and doctors during the ED visit. We collected both patient level characteristics and visit level characteristics. Patient level characteristics are characteristics that remain the same across visits, such as sex and comorbidities (including the presence of a progressive disease and the body system affected by the comorbidity with a separate group for a device in situ). Visit-level characteristics are specific for each visit and include age, time and date of arrival, type of referral, mode of arrival, triage classification, presenting problem category, severity category, additional examination, medication administered at the ED, vital signs, disposition after the ED visit and diagnosis category. In a subset of data (2%) we performed manually data comparison to check for misclassification. Also, data were checked for outliers, validity and completeness. A minimum anonymised database to reproduce the analyses is available through the S1 Dataset.

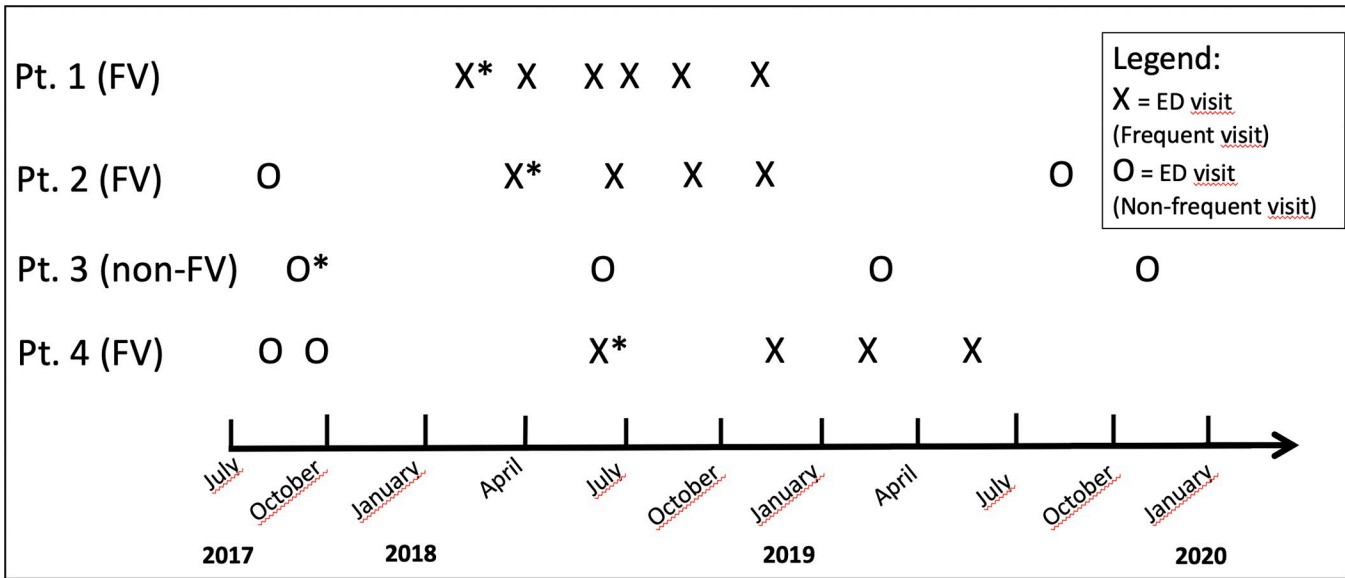

**Fig 1. Example of frequent visitors and frequent visits.** Index visits are marked with an * FV: Frequent visitor.

## Definition of characteristics

Age was evaluated as a continuous and categorical variable; other variables were categorized into clinically relevant categories. Specifically, vital signs were categorized based on the Advanced Pediatric Life Support guidelines [17]. Triage classification was based on the Manchester Triage System which divides patients into five urgency categories (immediate, very urgent, urgent, standard and non-urgent). For clinical relevance and given the low numbers in the 'immediate group' and non-urgent category, we combined immediate/very urgent and standard/non-urgent, resulting in a three-level triage urgency variable. Severity of an ED visit was determined based on type of management during the entire ED visit similar to the approach in the TrIAGE project [18, 19] (S1 Table), and included the levels not severe, severe and very severe.

Data export of International Classification of Diseases, Clinical Modification (ICD-10-CM) codes [20] linked to the ED visit were used to define new diagnoses (codes assigned at the date of the ED visit) and existing comorbidities (codes already assigned before the ED visit) for each patient. Comorbidity complexity was categorized in 3 levels of complexity using the Pediatric Medical Complexity Algorithm (PMCA) by Simon et al. [21]; children with complex chronic diseases (C-CD), children with non-complex chronic diseases (NC-CD) and children without chronic diseases (CD) [21] (S2 Table). To get more detail, comorbidities were also divided by body system involved. A separate group of patients with a device in situ was discriminated, except for patients with an external ventricular drain (EVD), who were included in the group neurologic comorbidity. Acute diagnoses were assigned in a 3-step approach. First, we used ICD-10 codes assigned at the day of visit. Second, we classified acute diagnoses based on information from presenting problem reported in the Manchester Triage System (MTS), the used MTS flowchart category or temperature [22]. Finally, patients with missing acute diagnoses from previous steps were categorized into the group other non-communicable disease. This approach yielded 65 categories that were further regrouped into 23 categories as used in previous literature [23] (S3 Table); these categories were used to describe the study population. For the final model, we reclassified these 23 categories into three main groups (communicable diagnosis, intoxication/injuries and non-communicable diagnosis). Also, we determined whether the acute diagnosis was corresponding to the body system affected by the comorbidity (S4 Table).

## Missing data

Multiple imputation was used for missing data on type of referral and admission as these missing values are assumed to be missing at random [24, 25]. The variables age, sex, moment of the day, moment of the week, referral category, arrival mode, diagnostic tests, problem category, triage category, medication, type of admission, comorbidity, vital signs and diagnose categories were included in the multiple imputation model to impute missing data for referral category and type of admission. This resulted in 10 complete databases, analyses were performed on these 10 databases and pooled estimates of the results of these analyses were used. Missing data in diagnostic procedures (imaging or laboratory tests) were considered as not performed, and missing data for vital signs were considered as not being deviating from normal. Children with missing data on the variable 'arrival mode' were assumed not arrived by ambulance.

## Data analysis

Descriptive analysis of characteristics of the study cohort at patient- and at visit-level were performed by frequencies and percentages for categorical variables and by means (standard

deviation) or medians (interquartile range) for continuous variables. We evaluated differences between FVs and non-FVs by chi-squared and student t-tests or Mann Whitney U-tests.

At patient-level we explored the reason for ED visit between children with and without comorbidity and compared FVs with non FV.

Subsequently, to determine predictors for a recurrent ED visit a Prentice Williams and Peterson gap time (PWP-GT) cox-based model was developed. In this model, the outcome variable is time until a return visit to the ED. This model accounts for clustering by patients, allowing us to include both patient-level and visit-level characteristics, and includes time at risk for a next visit. For this model, visits to the ED at the 30[th] of December 2020 were excluded as these children were not at risk to return to the ED. Predictors included in this study were based on their possible association with repeated ED visits and availability.

For all analyses a p-value of 0.05 or lower was considered statistically significant. Descriptive analyses were performed using the statistical package for the social sciences version 25 (SPSS Inc., Chigaco, Illinois, USA). Time-to-recurrent event analyses were performed using Stata/MP version 16 (StataCorp LLC, College Station, Texas).

## Results

During the study period, 19,186 visits to the ED of the Erasmus MC-Sophia Children's hospital were recorded. Due to a technical issue, data from a random sample of 2,789 visits (14.5%) could not be extracted with relevant data from the hospital informatics system (mostly due to lacking patient identifier codes), leaving 16,397 ED visits for inclusion. A total of 10,209 children included in this study visited the ED at least once within our study period. Visit rate ranged from 1 to 32 visits within our 3.5-year study period, with 7,709 (75.5%) having one visit, 1,347 (13.2%) having two visits, 463 (4.5%) having three visits and 690 (6.8%) children having four or more visits. In total, 500 (4.9%) out of the 10,209 children visited the ED four or more times within 365 days and are therefore called FVs. The frequent visits made by these children accounted for 3,481 (21.2%) of the total visits during the study period. Median number of visits among non-FVs was 1 (IQR 1–1), and for the FVs the median of visits was 6 (IQR 5–9) during the entire study period.

### Patient characteristics

Patient characteristics of FVs and non-FVs are presented in Table 1. At their index visit median age of FVs was younger compared with non-FVs. Overall, a higher rate of boys visiting the ED was found, and equally distributed among FVs and non-FVs (56.8% and 57.0% boys respectively). Most notably, FVs were more often children with comorbidities (77.8%) when compared to non-FVs (18.3%), in particular complex comorbidities (48.8% versus 12.7% respectively). Children with a neurologic comorbidity were most frequently present among all ED visitors (8.1%), and among the FVs (19.0%).

### Visit characteristics

Characteristics of the ED visits are presented in Table 2 and compared among frequent visits and non-frequent visits. Most visits to the ED were made during out of office hours in both groups, but frequent visits were more often made in the weekend. Frequent visits were also more often related to laboratory testing, admission and higher urgency, compared to non-frequent visits. Frequent visits were less often referred by either a GP or a specialist, arrived by ambulance, categorized as non-urgent/standard visits and imaging was less often done. In addition, frequent visits were less often related to intoxications or injuries (8.0%).

**Table 1. Patient characteristics non FVs and FVs.**

| | Frequent visitor | |
|---|---|---|
| | No (N = 9,709) | Yes (N = 500) |
| Age at index visit*, median (IQR) | 5.8 (1.7; 12.2) | 4.3 (1.1; 11.4) |
| Boys, %(*n*) | 56.8% (5,510) | 57.0% (285) |
| Comorbidity severity[1]*, %(*n*) | | |
| • No CD | 71.6% (6,955) | 22.2% (111) |
| • NC-CD | 15.6% (1,519) | 29.0% (145) |
| • C-CD | 12.7% (1,235) | 48.8% (244) |
| Body system comorbidity[1,2]*, %(*n*) | | |
| • No | 71.6% (6,955) | 22.2% (111) |
| • Neurologic | 7.6% (735) | 19.0% (95) |
| • Cardiac | 3.2% (315) | 6.2% (31) |
| • Pulmonal | 2.9% (286) | 8.2% (41) |
| • Musculo/skeletal | 2.5% (240) | 8.4% (42) |
| • Gastrointestinal | 2.5% (238) | 6.8% (34) |
| • Otologic | 2.4% (232) | 7.8% (39) |
| • Hematologic | 2.4% (230) | 7.8% (39) |
| • Malignancy | 1.6% (158) | 9.0% (45) |
| • Renal | 1.5% (148) | 8.0% (40) |
| • Device in situ | 1.5% (145) | 3.6% (18) |
| • Genetic | 1.5% (142) | 5.2% (26) |
| • Craniofacial | 1.5% (141) | 4.2% (21) |
| • Ophthalmologic | 1.5% (141) | 5.4% (27) |
| • Immunologic | 1.4% (134) | 4.4% (22) |
| • Metabolic | 1.1% (104) | 4.0% (20) |
| • Genital | 0.9% (88) | 2.0% (10) |
| • Mental Health | 0.7% (67) | 1.0% (5) |
| • Endocrinological | 0.6% (61) | 1.2% (6) |
| • Dermatologic | 0.0% (3) | 0.2% (1) |

IQR: inter quartile range; C-CD: complex chronic diseases; NC-CD: non-complex chronic diseases; CD: chronic diseases; PMCA: Pediatric Medical Complexity Algorithm.

[1]Based on the PMCA by Simon et al. [21].

[2]Because children can have more than one comorbidity columns do not add up to 100%.

Significant differences (p<0.05) are marked with an *.

## Reason for visiting the ED

Reasons for frequent visits were more often categorized as communicable and non-communicable conditions, compared to non-frequent visits (Table 2). Injuries and intoxications were rare among frequent visits. We explored the reasons for visits in subgroups of children based on type of comorbidity (S5 Table) and present the main comorbidity categories in Fig 2. Although children with malignancies and renal comorbidities comprise less ED visits in total, they are most often FVs (22.2% and 21.3% respectively). On the contrary, children with a cardiac or neurologic comorbidity comprised a large number of children but just a small group of FVs (9.0% and 11.4% respectively) (S5 Table). At patient level we determined the reasons for visiting the ED grouped by comorbidity's body system (Fig 2). This figure visualizes the relative contribution of communicable, non-communicable and trauma causes. Comparing the bars of FVs versus all visitors, it can be concluded that in all comorbidity types, we observed a

**Table 2. Visit characteristics frequent visits and non-frequent visits.**

| | | Frequent visit | |
|---|---|---|---|
| | | **No (n = 12,916)** | **Yes (n = 3,481)** |
| **Age** | median (IQR) | 5.9 (1.9; 12.1) | 5.2 (2.0; 12.0) |
| **Age category**[*] | | | |
| • < 1 year | % (n) | 16.5% (2,133) | 14.2% (494) |
| • 1–2 year | % (n) | 9.5% (1,229) | 10.6% (369) |
| • 2–5 year | % (n) | 19.3% (2,493) | 23.9% (831) |
| • 5–12 year | % (n) | 29.1% (3,763) | 26.0% (906) |
| • 12 year and older | % (n) | 25.5% (3,298) | 25.3% (881) |
| **Time of presentation** | % Out of office (n) | 61.1% (7,895) | 62.1% (2,160) |
| **Moment of the week**[*] | % Weekend (n) | 24.3% (3,145) | 27.8% (967) |
| **Time of day**[*] | | | |
| • Daytime | % (n) | 50.0% (6,453) | 52.1% (1,813) |
| • Evening | % (n) | 41.6% (5,375) | 38.6% (1,345) |
| • Night | % (n) | 8.4% (1,088) | 9.3% (323) |
| **Referral**[*] | | | |
| • Self-referral | % (n) | 60.8% (7,847) | 72.9% (2,538) |
| • GP | % (n) | 11.7% (1,505) | 4.6% (161) |
| • Specialist | % (n) | 5.6% (724) | 2.4% (83) |
| • Other | % (n) | 7.0% (905) | 1.3% (45) |
| • Missing | % (n) | 15.0% (1,935) | 18.8% (654) |
| **Arrival mode**[*] | | | |
| • Own transportation | % (n) | 62.0% (8,004) | 67.0% (2,333) |
| • Ambulance | % (n) | 20.7% (2,674) | 11.7% (406) |
| • Other | % (n) | 0.1% (11) | - |
| • Missing | % (n) | 17.2% (2,227) | 21.3% (742) |
| **Triage category**[*] | | | |
| • Standard/non-urgent | % (n) | 41.6% (5,372) | 36.5% (1,269) |
| • Urgent | % (n) | 41.1% (5,307) | 46.9% (1,633) |
| • Immediate/very urgent | % (n) | 17.3% (2,237) | 16.6% (579) |
| **Problem category**[*] | | | |
| • Trauma | % (n) | 27.4% (3,542) | 5.6% (195) |
| • Medical/psychological | % (n) | 72.6% (9,374) | 94.4% (3,286) |
| **Severity** [*][1] | | | |
| • Not severe | % (n) | 73.3% (9,471) | 69.0% (2,403) |
| • Severe | % (n) | 23.3% (3,008) | 28.4% (990) |
| • Very severe | % (n) | 3.4% (437) | 2.5% (88) |
| **Diagnostic test**[*][2] | | | |
| • Laboratory | % (n) | 15.2% (1,961) | 24.7% (861) |
| • Imaging | % (n) | 29.9% (3,867) | 20.4% (711) |
| **Medication**[*] | % Yes (n) | 6.0% (772) | 8.1% (283) |
| • Emergency medication | % Yes (n) | 0.3% (33) | 0.2% (7) |
| **Admission**[*] | | | |
| • No admission | % (n) | 61.7% (7,975) | 53.3% (1,854) |
| • Short admission | % (n) | 1.9% (240) | 1.9% (67) |
| • Admission | % (n) | 20.3% (2,621) | 24.8% (862) |
| • PICU | % (n) | 3.1% (394) | 2.3% (80) |
| • Missing | % (n) | 13.1% (1,686) | 17.8% (618) |
| **Diagnose category**[*][2] | | | |

*(Continued)*

**Table 2.** (Continued)

| | | Frequent visit | |
|---|---|---|---|
| | | No (n = 12,916) | Yes (n = 3,481) |
| • **Communicable** | % (n) | 27.4% (3,541) | 41.3% (1,436) |
| • Unspecified | % (n) | 13.5% (1,742) | 20.6% (718) |
| • GIT | % (n) | 4.1% (536) | 9.7% (337) |
| • Respirator | % (n) | 2.8% (360) | 4.5% (155) |
| • ENT | % (n) | 4.4% (565) | 3.2% (113) |
| • Other specified[3] | % (n) | 3.3% (426) | 3.9% (137) |
| • **Intoxication/Injury** | % (n) | 34.1% (4,409) | 8.0% (280) |
| • **Non-communicable** | % (n) | 47.8% (6,169) | 61.2% (2,132) |
| • Unspecified | % (n) | 15.7% (2,031) | 20.9% (729) |
| • GIT | % (n) | 8.3% (1,066) | 8.4% (293) |
| • Neurologic | % (n) | 6.8% (882) | 9.3% (322) |
| • Respirator | % (n) | 5.0% (641) | 9.0% (315) |
| • Other specified[4] | % (n) | 14.9% (1,922) | 17.8% (620) |
| **Diagnose corresponding to comorbidity[5]\*** | % Yes (n) | 13.6% (1,752) | 26.0% (906) |

Significant differences (p<0.05) are marked with an *.

[1]Severity was determined based on information of the entire ED visit [18] (S1 Table).

[2]Because children could have more than one diagnoses category and diagnostic test columns do not add up to 100%.

[3] Other specified communicable diagnoses categories included: eye infections, skin infections and urinary infections.

[4] Other specified non-communicable diagnose categories included: congenital malformations, neoplasms, circulatory, urogenital, eye/ear, hematologic, psychologic, skin, perinatal, endocrinologic and muscle/joint.

[5]Diagnosis corresponding to comorbidity are presented in S4 Table,

GP: General Practitioner; PICU: Pediatric Intensive Care Unit; GIT: gastro-intestinal; ENT: ear nose and throat.

higher contribution of non-communicable conditions, followed by communicable conditions and only a small proportion of injuries/intoxications. In 13 out of 17 comorbidity types, communicable diagnoses were more common in FVs when compared to non FVs, although in both groups non-communicable conditions were the most common reason for visiting. Intoxications/injuries were relatively common reasons to visit the ED in children with hematologic comorbidity (for both FVs and non FVs) in particular. In the group of children without comorbidity, a dominance of non-communicable diagnosis was observed, with intoxications/injuries being second. Similar to children with comorbidities, in the subgroup of FVs without comorbidity the contribution of communicable diagnosis was substantially higher (23.4% for non FVs versus 44.0% for FVs). More specific, non-communicable diagnoses were most often related to the body system affected by the comorbidity in 12 out of 17. For the comorbidities affecting the gastrointestinal and urogenital tract communicable diagnoses were mostly related to the body system involved, but for the respiratory group this was less clear. For the other body systems involved, communicable diagnoses were most often classified into 'unspecified origin' (S5 Table).

### Risk factors for a return visit

This analysis of 16,386 available visits identified age (<1 year) (HR 1.17, 95% CI 1.07–1.29), being referred by a specialist (HR 1.11, 95% CI 1.01–1.21), being triaged as urgent (HR 1.13, 95% CI 1.06–1.20) receiving medication at the ED (HR 1.16, 95% CI 1.04–1.30), and the presence of a comorbidity (non-complex: HR 1.66, 95% CI 1.52–1.81; complex: HR 2.00, 95% CI

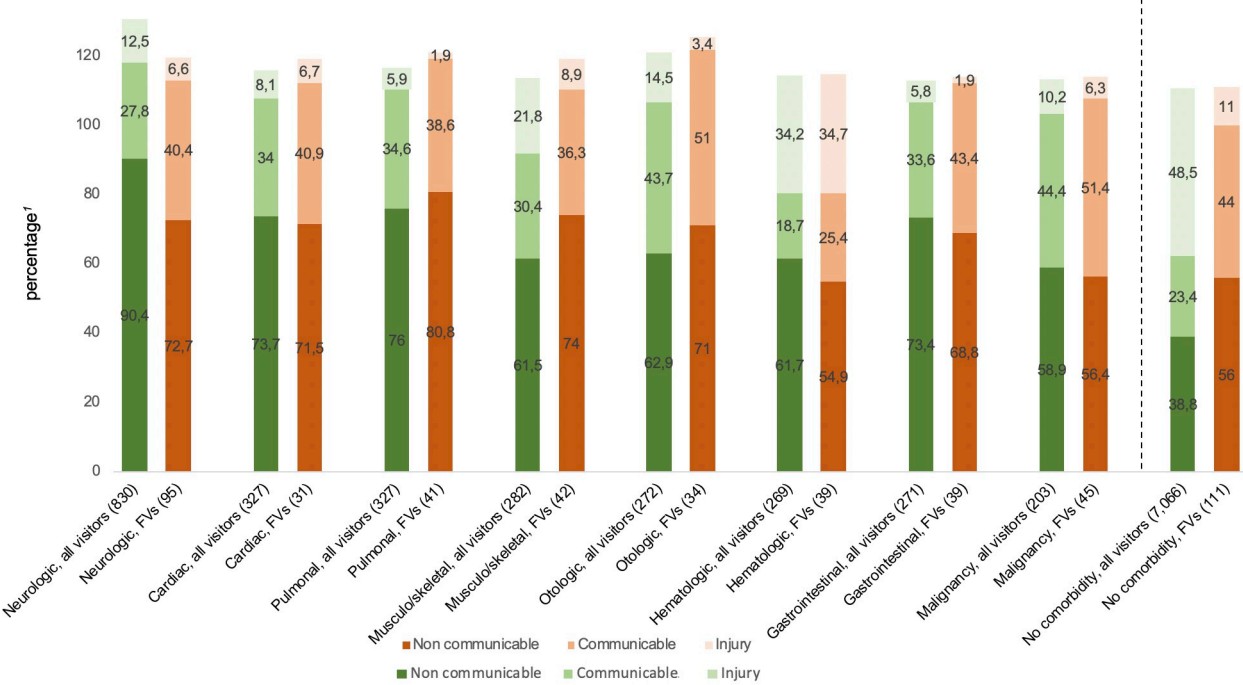

**Fig 2. Reason to visit the ED for the main comorbidity categories.** [1]Because children could have more than one diagnoses category columns do not add up to 100%. ED, Emergency Department.

1.84–2.16) as most important determinants for a return visit (Table 3). A HR of 1.17 means that children presenting at the ED below the age of 1 year old have 17% more hazard for a recurrent visit compared to children aged 12 and over. Both arriving by ambulance and having an intoxication or injury lowers the hazard for a return visit.

## Discussion

### Findings

In this study on pediatric ED visits among an inner-city university hospital in the Netherlands, we observed that frequent visits accounted for a large percentage of the total pediatric ED visits (21.3%). FVs were slightly younger at index visit and more often suffered from comorbidities compared to non-FVs. When comparing the visit characteristics, frequent visits were less often characterized by referral from a GP or specialist or arrival by ambulance. However, these visits were less often non urgent. Frequent visits were more often for medical problems than trauma. Although overall the percentage of communicable conditions was lower compared to non-communicable conditions, communicable conditions have a more prominent role in FVs, in both children with and without comorbidities. The highest hazard for a return visit were observed for the presence of non-complex and especially complex comorbidities. Other risk factors for a recurrent visit include being less than 1 year of age, being referred by a specialist, being triaged as urgent and receiving medication at the ED.

### Comparison to previous literature

Similar to our study, previous studies in children found that FVs were more often children with CDs [11]. Also similar to a previous study in adults, we observed slightly higher urgency

**Table 3. Prentice Williams and Peterson gap time (PWP-GT) cox-based model for a recurrent ED visit.**

| | HR | 95% CI | *p*-value |
|---|---|---|---|
| **Age category** | | | |
| • < 1 year | **1.17** | **1.07–1.29** | **<0.01** |
| • 1–2 year | 1.04 | 0.94–1.16 | 0.45 |
| • 2–5 year | 1.08 | 1.00–1.18 | 0.07 |
| • 5–12 year | 1.04 | 0.96–1.13 | 0.36 |
| • 12 year and older | *reference* | *reference* | *reference* |
| **Moment of the week** (weekend) | 0.96 | 0.91–1.02 | 0.18 |
| **Time of day** | | | |
| • Daytime | *reference* | *reference* | *reference* |
| • Evening | 0.99 | 0.94–1.05 | 0.75 |
| • Night | 0.93 | 0.84–1.02 | 0.12 |
| **Referral** | | | |
| • Self-referral | *reference* | *reference* | *reference* |
| • GP | 0.95 | 0.88–1.02 | 0.17 |
| • Specialist | **1.11** | **1.01–1.21** | **0.03** |
| • Other | **0.73** | **0.62–0.86** | **<0.01** |
| **Arrival mode** (ambulance) | **0.82** | **0.76–0.90** | **<0.01** |
| **Triage category** | | | |
| • Standard/non-urgent | *reference* | *reference* | *reference* |
| • Urgent | **1.13** | **1.06–1.20** | **<0.01** |
| • Immediate/very urgent | 1.06 | 0.97–1.15 | 0.20 |
| **Diagnostic test** | | | |
| • Laboratory | 1.06 | 0.99–1.13 | 0.09 |
| • Imaging | 0.97 | 0.91–1.04 | 0.41 |
| **Diagnose category** | | | |
| • Infectious (yes) | 0.99 | 0.90–1.09 | 0.87 |
| • Intoxication/Injury (yes) | **0.56** | **0.50–0.64** | **<0.01** |
| • Other (yes) | 1.00 | 0.91–1.10 | 0.98 |
| **Diagnose corresponding to comorbidity[1] (yes)** | 0.93 | 0.87–1.00 | 0.05 |
| **Medication (yes)** | **1.16** | **1.04–1.30** | **0.01** |
| **Admission** | | | |
| • No admission | *reference* | *reference* | *reference* |
| • Short admission | 0.96 | 0.82–1.12 | 0.59 |
| • Admission | 0.95 | 0.89–1.01 | 0.11 |
| • PICU | 0.88 | 0.77–1.02 | 0.08 |
| **Comorbidity severity[2]** | | | |
| • No CD | *reference* | *reference* | *reference* |
| • NC-CD | **1.66** | **1.52–1.81** | **<0.01** |
| • C-CD | **2.00** | **1.84–2.16** | **<0.01** |

[1]Diagnosis corresponding to comorbidity are presented in S4 Table.

[2]Based on the PMCA by Simon et al. [21].

HR: Hazard Ratio; CI: Confidence Interval; GP General Practitioner; CD: Chronic Disease; NC-CD: Non-Complex Chronic Disease; C-CD: Complex Chronic Disease.

classification among FVs when compared to non FVs [9]. Also, children and adult FVs are both more likely to be admitted to the hospital. Therefore, as concluded in the study on adult ED FVs, in our study pediatric FVs seem to visit the ED when requiring specialized medical care [9].

In an international comparison we observed similar FVs and a slightly higher visit rate [3, 4, 7, 9, 10, 13, 26]. Although the approach of 4 visits counted from an index visit is in line with most studies, variation in definitions may contribute to the observed differences. Unlike a previous study [10], we did not see high rates for ENT and respiratory problems as reason to visit the ED for frequent visits. In our study, the main reasons for frequent visits were diverse.

## Strengths

This study includes data over a time period of 3.5 years, with a large number of FVs. This may be related to the fact that our ED has adherence area for 25% of the Netherlands pediatric tertiary care. Although our study is a single center study, our hospital's inner-city function results in a multicultural and socially diverse patient population similar to other EDs in inner cities. In addition to previous studies we performed time series analysis. By doing this we were able to combine patient and visit characteristics and account for recurrent visits within a patient. Also, the high number of patients and revisits (16,386) allowed us to analyze a large number of potential determinants.

## Limitations

First, with the use of administrative data there is a chance of misclassification and missing data. However, using administrative data is the most efficient approach to perform this study with a high number of cases included. Manual data comparison did not detect substantial flaws. For the variables with missing values we dealt with the missing data in appropriate manners by using multiple imputation. Second, we did not have information on scheduled outpatient visits and socio-economic status which has previously been reported to influence revisits [10, 27]. Last, we did not have information on visits to other EDs. This may have caused underestimation of the proportion FVs, as children may have presented at different EDs [28]. Because children with a comorbidity are considered regular visitors of the same hospital, we think this would mainly affect non-comorbid children.

## Implication for future research and clinical practice

Our observations that FVs are mostly related to comorbid conditions may be as we could have expected from a clinical perspective on beforehand. However, we now have proven it in a large population. Next, we show that causes of frequent ED consultation are not always related to the underlying comorbidity itself, but frequently relate to unspecified infectious causes. It is important to note that frequent visits were not less often triaged as (very) urgent or admitted when compared to non-frequent visits. This suggests but does not prove that FVs visit the ED for appropriate reasons. Therefore, rather than discouraging FVs to visit the ED, this calls for strategies to channel ED visits to more appropriate healthcare pathways. An ED consult is the most expensive form of health care compared to a consult with a specialist or a GP [6]. Given the dominance of communicable conditions, preventive measures or instructions for care at home to reduce infections may be considered. Similarly, for hematologic patients in particular, a special (fast track) healthcare pathway for injuries might be useful. Addressing risk factors for specific complex comorbidities may provide more comprehensive information for FVs within different subgroups (e.g. type of comorbidity) but was beyond the scope of this study given the relative low numbers.

Furthermore, it is important to include the opinion and expectations of children frequently visiting the ED and their parents. To reduce their ED visits, it is important to know if patients had contact with a healthcare provider prior to their ED visit. A designated contact for

children and their parents might help channeling them towards more regular scheduled health care, particularly for children with comorbidities.

## Conclusion

This large study based on electronic data extraction confirms that pediatric FVs of the ED account for a high proportion of visits. The strongest risk factor for a child to return to the ED is the presence of (multiple) comorbidities. Also, in children with comorbidities reason for visit is often an infection or relates to the body system involved by their comorbidity. Identified risk factors may guide ways to optimize patient flows and patient outcomes e.g. by restructuring healthcare pathways. Health care pathways, including safety-netting strategies for acute manifestations from their comorbidity, or for infectious conditions in general may contribute to support parents and redirect some patients from the ED.

## Supporting information

**S1 Dataset.**
(XLSX)

**S1 Table. True severity classification [18].** * Haemodynamic support (e.g. significant IV fluid in case of hypotension, blood administration or control of major bleeding) or emergency medications (e.g. atropine, adenosine, inotropics, epinephtrine, nalaxon).
(PDF)

**S2 Table. Pediatric Medical Complexity Algorithm [21].** NA not applicable, C-CD Complex Chronic Disease, NC-CD Non-Complex Chronic Disease, CD Chronic Disease. *The examples used in this document to illustrate definitions of medical complexity are intended to demonstrate characteristics specified in the definition/descriptions. It is not our intention to imply that specific diseases and conditions are by default linked to the categories that they were used to illustrate.
(PDF)

**S3 Table. Division of acute diagnosis.** C Communicable NC Non-Communicable.
(PDF)

**S4 Table. Diagnose corresponding to comorbidity.** C Communicable NC Non-Communicable.
(PDF)

**S5 Table. Reason for visiting the ED split by comorbidity at patient level.** *Because children could have more than one diagnoses category rows do not add up to 100%. Diagnoses corresponding to the body system of the comorbidity are marked bold. NC: Non-Communicable; I: Intoxication/injuries; C: Communicable.
(PDF)

## Author Contributions

**Conceptualization:** Sanne E. W. Vrijlandt, Joany M. Zachariasse, Rianne Oostenbrink.

**Data curation:** Sanne E. W. Vrijlandt.

**Formal analysis:** Sanne E. W. Vrijlandt, Daan Nieboer, Rianne Oostenbrink.

**Methodology:** Sanne E. W. Vrijlandt, Daan Nieboer, Rianne Oostenbrink.

**Project administration:** Rianne Oostenbrink.

**Resources:** Rianne Oostenbrink.

**Supervision:** Rianne Oostenbrink.

**Visualization:** Joany M. Zachariasse.

**Writing – original draft:** Sanne E. W. Vrijlandt.

**Writing – review & editing:** Sanne E. W. Vrijlandt, Daan Nieboer, Joany M. Zachariasse, Rianne Oostenbrink.

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
