## [Decision Letter · Decision Letter 0]

20 Oct 2021

PONE-D-21-20440Characteristics of pediatric emergency department frequent visitors and their risk of a return visit: a large observational study using electronic health record data.PLOS ONE

Dear Dr. Oostenbrink,

Thank you for submitting your manuscript to PLOS ONE. After careful consideration, we feel that it has merit but does not fully meet PLOS ONE’s publication criteria as it currently stands. Therefore, we invite you to submit a revised version of the manuscript that addresses the points raised during the review process.

We look forward to receiving your revised manuscript.

Kind regards,

Biswadev Mitra, MBBS, MHSM, PhD, FACEM

Academic Editor

PLOS ONE

Additional Editor Comments:

Dear authors,

This is an important topic for research and incorporates a population that has not been researched frequently.

The methodology is well-explained and the results are easy to follow.

I have some minor comments only:

Introduction

1. Best not to assume that patients with FV have unfulfilled medical needs. Please consider re-phrasing this sentence.

2. Last sentence, "Which provides..." is not grammatically correct.

Results

3. There are many patients under the category of "other" in FV. Is it possible to provide a more detailed description of this subgroup?

4. Ln 266. Please reword "..you may conclude.." to we may or it can be concluded

5. Under risk factors, best not to present any interpretation of results here.

6. Under conclusions, I am not convinced that this study ahs indicated re-structuring of pathways. Suggest deleting it from the conclusion or re-wording

Reviewers' comments:

Reviewer's Responses to Questions

**Comments to the Author**

1. Is the manuscript technically sound, and do the data support the conclusions?

Reviewer #1: Yes

2. Has the statistical analysis been performed appropriately and rigorously? 

Reviewer #1: I Don't Know

3. Have the authors made all data underlying the findings in their manuscript fully available?

Reviewer #1: No

4. Is the manuscript presented in an intelligible fashion and written in standard English?

Reviewer #1: No

5. Review Comments to the Author

Reviewer #1: This study analyses data from a hospital centre serving a large urban population in order to characterise the factors most likely to be associated with frequent ED attendance. The tertiary care provision is for a larger area. One of the strengths is its analyses by person and by visit type. The categorisation of conditions uses a number of other study methods which gives some consistency across similar studies.

The data was incomplete in a round 15% of cases and it would be useful to know if there were any systematic biases noted ( age, severity etc. compared to the included group).

I would want a statistics expert to comment on the choice of tests used and their appropriateness.

36% of the visits classified as frequent attenders were in the non-urgent category and a cross tabulation against complexity would aid interpretation as other studies of large hospital populations have shown this to be possibly related to access and other demographic issues ( ethnicity, residency location etc) . It would strengthen the conclusion made that "frequent visits were not less severe than compared to non FVs"

There were a number of grammatical errors which I list below :-

Suggest L99 changed to read "As FVs have some sort of unfulfilled medical or social need , insight...."

L149 replace "diagnose" with "diagnosis"

L209 replace "allows" with "allowing us"

L290 replace "hazard" with "risk"

similarly on line 307

L351 replace with phrase "have expected from a clinical perspective."

L352 replace " is" with "are"

L353 replace "relates" with "relate"

L376 suggest rephrasing This study suggest that there may be a case for restructuring health care pathways for children with co-morbidities in order to optimise patient flows....."

L379 rephrase "...…contributing to supporting parents and redirecting some patients from the ED"

6. PLOS authors have the option to publish the peer review history of their article (what does this mean?). If published, this will include your full peer review and any attached files.

Reviewer #1: **Yes: **Professor Mitch Blair

---

## [Author Response · Author response to Decision Letter 0]

17 Dec 2021

Editor’s comments:

Comment 1. Best not to assume that patients with FV have unfulfilled medical needs. Please consider re-phrasing this sentence.

Answer 1. Thank you for this suggestion, we have adjusted this sentence (Line 105).

‘’..as it might be possible to fulfill their medical needs in other healthcare pathways.’’

Comment 2. Last sentence, "Which provides..." is not grammatically correct.

Answer 2. We have re-phrased this sentence (Line 119).

‘’This contributes to …’’

Comment 3. There are many patients under the category of "other" in FV. Is it possible to provide a more detailed description of this subgroup?

Answer 3. We had chosen to group the smallest groups into a category ‘others’, in order to avoid the table being too long. However, we have now adjusted the table (table 1 page 9) to show all categories. 

Comment 4. Ln 266. Please reword "..you may conclude.." to we may or it can be concluded

Answer 4. We have changed this (Line 275) to ‘’..it can be concluded..’’.

Comment 5. Under risk factors, best not to present any interpretation of results here.

Answer 5. The sentence was intended to guide the reader how to understand the meaning of the figures rather than to reflect the interpretation of the authors, we have re-phrased the sentence slightly and removed the term ‘interpreted” (Line 301) ‘’A HR of 1.17 means…’’.

However, when the editor considers it redundance we can leave it out.

Comment 6. Under conclusions, I am not convinced that this study has indicated re-structuring of pathways. Suggest deleting it from the conclusion or re-wording

Answer 6. We understand that the editor raises this issue and we have therefor re-worded this sentence (Line 388) ‘’Identified risk factors…’’. Accordingly, we also changed the conclusion in the abstract (line 49) ‘’Healthcare pathways, including…”.

Reviewer’s comments: 

Comment 1. The data was incomplete in a round 15% of cases and it would be useful to know if there were any systematic biases noted ( age, severity etc. compared to the included group). I would want a statistics expert to comment on the choice of tests used and their appropriateness.

Answer 1. We understand this concern, and indeed, we had missings for some variables, however as shown in table 2 distribution of missing was similar among frequent visits and non-frequents visits (our outcome). In the final model only two variables with missings were included, although these missings were multiple imputed to make all data available for analysis. As we assumed the comment of the statistics experts of choice of tests was related to this remark, we added in the manuscript a reference on dealing with missings (methods section page 7, Line 197, ref 24 and 25). A statistician was involved in the analyses and has co-authored this paper, and reviewed the reviewer’s comments and the revised version (author D. Nieboer).

Comment 2. 36% of the visits classified as frequent attenders were in the non-urgent category and a cross tabulation against complexity would aid interpretation as other studies of large hospital populations have shown this to be possibly related to access and other demographic issues (ethnicity, residency location etc). It would strengthen the conclusion made that "frequent visits were not less severe than compared to non FVs".

Answer 2. We do not fully understand whether you would like a cross tabulation of comorbidity complexity or urgency severity against triage category. We assume you mean severity instead of (comorbidity) complexity and cross tabulated this against triage category for frequent visits as well as for non-frequent visits. This showed no difference between the frequent visits and non-frequent visits (supplemental data file 1). However, we have changed the sentence to the actual data (Line 362) ‘’not less often triaged as…’’.

Comment 3 There were a number of grammatical errors which I list below :

Suggest L99 changed to read "As FVs have some sort of unfulfilled medical or social need , insight....", L149 replace "diagnose" with "diagnosis", L209 replace "allows" with "allowing us", L290 replace "hazard" with "risk" similarly on line 307, L351 replace with phrase "have expected from a clinical perspective.", L352 replace " is" with "are", L353 replace "relates" with "relate", L376 suggest rephrasing This study suggest that there may be a case for restructuring health care pathways for children with co-morbidities in order to optimise patient flows.....", L379 rephrase "...…contributing to supporting parents and redirecting some patients from the ED"

Answer 3. Thank you for the suggestions we have corrected the grammatical errors. 

We only preferred to keep the term “hazard” (Line 302 and 304) in the results section, to stay with the proper meaning of the used analysis, although it can be used as ‘risk’ in a more interpretative way, as used in the discussion (line 319). However, if the editor or reviewer consider these terms interchangeable, we are happy to change this.

---

## [Editor Report · Decision Letter 1]

23 Dec 2021

Characteristics of pediatric emergency department frequent visitors and their risk of a return visit: a large observational study using electronic health record data.

PONE-D-21-20440R1

Dear Dr. Oostenbrink,

We’re pleased to inform you that your manuscript has been judged scientifically suitable for publication and will be formally accepted for publication once it meets all outstanding technical requirements.

Kind regards,

Biswadev Mitra, MBBS, MHSM, PhD, FACEM

Academic Editor

PLOS ONE
---

## [Editor Report · Acceptance letter]

6 Jan 2022

PONE-D-21-20440R1 

Characteristics of pediatric emergency department frequent visitors and their risk of a return visit: a large observational study using electronic health record data. 

Dear Dr. Oostenbrink:

I'm pleased to inform you that your manuscript has been deemed suitable for publication in PLOS ONE. Congratulations! Your manuscript is now with our production department. 

Kind regards, 

on behalf of

Prof. Biswadev Mitra 

Academic Editor

PLOS ONE